# Knowledge, Attitudes, and Practices Regarding Antibiotic Sales in Pharmacies in Medellín, Colombia 2023

**DOI:** 10.3390/antibiotics12091456

**Published:** 2023-09-19

**Authors:** Daniel Ricardo Montes Colonia, Daniela Ramírez Patiño, Luis Felipe Higuita-Gutiérrez

**Affiliations:** 1Facultad de Medicina, Universidad Cooperativa de Colombia, Medellín 050012, Colombia; daniel.montesc@campusucc.edu.co; 2Hospital Pablo Tobón Uribe, Medellín 050034, Colombia; dany.rp24@gmail.com; 3Escuela de Microbiología, Universidad de Antioquia, Medellín 050010, Colombia

**Keywords:** antibiotics, community pharmacies, health knowledge, attitude, practice, prescription drug overuse

## Abstract

Objective: Describe the knowledge, attitudes, and practices regarding the sale of antibiotics in pharmacies in Medellín, Colombia. Method: A cross-sectional descriptive study was conducted in 277 selected pharmacies using a stratified sampling method with proportional allocation to represent all areas of the city. Knowledge, attitudes, and practices (KAPs) were assessed using a scale, analyzed with absolute and relative frequencies for each item, and represented in a global score ranging from 0 to 100, with a higher score indicating better KAPs. Data were analyzed using relative frequencies with 95% confidence intervals, the Mann-Whitney U test, the Kruskal-Wallis test, and linear regression. Results: Of the included pharmacies, 52.6% were chain pharmacies, 48.4% were attended by pharmacy assistants, and 59% of pharmacists had more than 5 years of experience. The median knowledge score was 70.8 (IQR 58.3–87.5), with 35.3% of pharmacists believing that antibiotics are effective in treating the common cold, 35.2% for treating COVID-19, and 29.4% considering them available for sale without a medical prescription. The attitude score was 53.3 (40.0–66.7), with 60.9% agreeing that prohibiting the sale of antibiotics without a prescription would decrease their sales. The practice score was 62.5 (40.0–79.2), with 65.4% of pharmacists stating that they sometimes sell antibiotics without a prescription due to patients struggling to obtain a medical consultation, 61.3% admitting to selling antibiotics without a prescription for urinary tract infections, and 41.3% for upper respiratory tract infections. Practices were predominantly influenced by pharmacy type (chain or independent) and, to a lesser extent, by knowledge and attitudes. Conclusion: Pharmacists in Medellín exhibit inadequate knowledge, attitudes, and practices regarding the use and sale of antibiotics without a medical prescription. These findings align with international evidence highlighting the need for educational and regulatory strategies promoting rational antibiotic use in pharmacies.

## 1. Introduction

Antibiotics are non-renewable resources. From an evolutionary perspective, their usage inevitably leads to the emergence of resistance. The irrational use of antibiotics results in the loss of their effectiveness and a limited availability of resources for treating future infections [1]. In fact, the World Health Organization (WHO) has warned that common infections may become potentially lethal diseases in the short term due to the absence of effective medications to treat them [2]. Despite this, antibiotic consumption in humans is increasing worldwide. Between the years 2000 and 2015, antibiotic consumption, measured in defined daily doses (DDD), increased by 65% (21.1–34.8 billion DDD), and the antibiotic consumption rate rose by 39% (11.3–15.7 DDD per 1000 inhabitants per day) [3]. The rise in usage leads to an increase in resistance, posing a public health problem at all levels due to the morbidity, mortality, disability, and costs it generates. It even endangers the progress made in sensitive areas, such as infant mortality [4]. Some estimates indicate that 56,524 newborns die each year in India due to neonatal sepsis attributed to antibiotic-resistant bacterial infections [5].

The inappropriate use and overuse of antibiotics in the agricultural, veterinary, and medical sectors constitute one of the main determinants of bacterial resistance [6]. However, globally, nearly half of antibiotics are acquired in retail pharmacies without a medical prescription [7], making pharmacies a key player in this issue. Retail pharmacies are often the first point of contact for patients seeking medical care, and it has been noted that over-the-counter antibiotics are primarily provided in these establishments upon patient request or based on the pharmacist’s advice [7,8,9].

Various authors have taken an interest in highlighting the contribution of pharmacists to the inappropriate use of antibiotics. A study conducted in Hungary found that 25% of pharmacists admitted to dispensing antibiotics without a medical prescription [10]. Another study conducted in 9 provinces of Sri Lanka (Asia) revealed that antibiotics were dispensed without a prescription in 1 out of every 3 pharmacies [11]. In Punjab, Pakistan, an evaluation of 573 pharmacies showed that 81.5% reported providing antibiotics without a prescription, despite 69.3% acknowledging that this contributed to antibiotic resistance [12]. In Brazil, a study involving pharmacists found that none of them possessed adequate knowledge about appropriate antibiotic use [13].

In Colombia, bacterial resistance is a significant topic in public health, as is evident in the national action plan to combat antimicrobial resistance [14]. However, studies have largely focused on hospital settings and the molecular mechanisms by which infectious agents develop resistance, overlooking community aspects of antibiotic misuse and abuse, particularly within the pharmaceutical sector. Specifically, in the city of Medellín, the second-largest city in Colombia, our research group has examined the knowledge, attitudes, and practices regarding antibiotics and their resistance among medical students [15] and the general population [16]. In both cases, inadequate knowledge, attitudes, and practices were identified. Given these circumstances, our current research aimed to describe the knowledge, attitudes, and dispensing practices of antibiotics among pharmacists in the city.

## 2. Results

A total of 277 pharmacies were included in the study, proportionally distributed across all areas of the city, with 52.6% being chain pharmacies. Those attending to the pharmacies were predominantly women (61.7%), with 48.4% holding the title of pharmacy assistant. Furthermore, 59% had more than 5 years of work experience, 66.9% believed that their education regarding antibiotics was adequate, and 22% reported selling more than 50 antibiotic tablets per day (Table 1). Each pharmacist was asked to list the three best-selling antibiotics in their pharmacy. The first one was amoxicillin, with 71.5% mentioning it as the top seller. The second most-sold antibiotic was azithromycin, mentioned by 35.4% of respondents, and the third was cephalexin, mentioned by 24.5%.

### 2.1. Knowledge

The median knowledge score was 70.8 (IQR 58.3–87.5) points; however, 35.3% indicated some degree of agreement that antibiotics are effective in treating the common cold, 35.2% believed that they are useful for treating COVID-19, 32.2% agreed that antibiotics can directly reduce fever, and 29.4% stated that antibiotics can be taken without a medical prescription (Figure 1). Knowledge levels were higher among pharmacy technologists, pharmacies located in the northwestern region, and those with more than 5 years of work experience (Table 2).

### 2.2. Attitudes

The attitudes index score was 53.3 (40.0–66.7). A total of 89.1% indicated some degree of agreement that, if they refused to sell antibiotics to a patient who did not need them, the patient could easily obtain them from another pharmacy. Furthermore, 84.9% believed that the sale of antibiotics without a medical prescription should be more closely controlled. However, 60.9% affirmed that banning the sale of antibiotics without a medical prescription in pharmacies would decrease their sales. Lastly, 77.8% agreed that the medical prescriptions received at their pharmacy contain errors and should be more closely monitored (Figure 2). In terms of attitudes, all scores were statistically similar (Table 2).

### 2.3. Practices

The practices index score was 62.5 (40.0–79.2). Within this category, 65.4% indicated that they have always, or almost always, sold antibiotics without a medical prescription because patients face difficulties in accessing a medical consultation. Additionally, 61.3% have sold antibiotics without a prescription to patients with urinary tract infections, 41.3% to patients with upper respiratory tract infections, 35.8% to patients with dental infections such as abscesses, and 37.2% to patients with undiagnosed infections (Figure 3). The practices index score was significantly lower (*p*-value 0.007) in retail pharmacies (Table 2).

### 2.4. Factors Associated with KAPs

Linear regression models were employed to identify factors associated with each index and their interrelationship. In each model, variables with *p*-values < 0.25 from bivariate analysis were included. It was found that knowledge is associated with the pharmacy’s location zone (β 1.429), the educational background of the attending personnel (β 4.207), and their work experience (β 3.672), while attitudes are solely associated with knowledge (β 0.127). Practices are influenced by both knowledge (β 0.211) and attitudes (β 0.310), but also, to a larger extent, by the type of pharmacy (β 5.763), indicating that working in chain pharmacies increases the practice index score by 5.763 (Figure 4).

## 3. Discussion

This study found that, in nearly one-third of pharmacies, there were misconceptions about the effectiveness of antibiotics for treating the common cold and COVID-19. As a result, a high frequency of antibiotic sales without a medical prescription was reported for upper respiratory tract infections. This finding is consistent with recent systematic reviews that highlight upper respiratory tract infections as one of the most common reasons for the non-prescription sale of antibiotics [17,18]. This pattern is also observed among physicians in the United States, where up to 10 million antibiotic prescriptions per year are inappropriately directed toward upper respiratory tract infections [19]. Such practice has been reported in patients as well. The source of this belief may stem from the fact that upper respiratory tract infections are often self-limiting; thus, the relief of symptoms while taking antibiotics is often interpreted as an indicator of their efficacy [20]. Disrupting this misconception requires interventions involving the participation of all stakeholders: patients, physicians, and pharmacists. In this regard, the study by Kandeel et al. presents significant insights [21]. Interventions within the city should follow a similar course of action.

Furthermore, 29.4% of pharmacists are not aware that antibiotics should only be sold with a medical prescription, despite the fact that all antibiotics in the country bear labels indicating them to be “prescription-only medication”. The sale of antibiotics without a medical prescription is prohibited in most countries worldwide [22]; however, non-compliance with regulations is a recurring issue. A systematic literature review encompassing 15 studies from 10 different countries evaluated various strategies to enforce regulations prohibiting the non-prescription sale of antibiotics. Common interventions included media campaigns, strengthening government inspections, retaining medical prescriptions, and educating pharmacists. Results were quite heterogeneous across studies; nevertheless, interventions seemed more effective when implemented collectively as a coordinated effort, involving pharmacist engagement in the intervention design [23]. It is advisable to take these prior experiences into account when devising actions for implementation within the city.

In the domain of attitudes, the lowest score was obtained. In nine out of ten pharmacies, it is believed that if they refuse to sell antibiotics to a patient who doesn’t need them, the patient could easily obtain them from another pharmacy. Similarly, 6 out of 10 indicated that if the sale of antibiotics without a medical prescription were prohibited, it would decrease their sales. This result unveils a lack of trust in colleagues’ compliance with the prescription-only sale norm of antibiotics and the financial benefits for pharmacies engaging in such practices. For these reasons, bacterial resistance has been conceptualized by some authors as a collective action problem. Collective action problems are situations where societal benefits are greater when everyone cooperates, while individual benefits are higher if they refrain from cooperating [24]. In this case, pharmacists have little to no reason to consider their contribution to bacterial resistance when making decisions about selling antibiotics without a medical prescription. However, they do perceive sales and profit reduction if they choose not to do so. In other collective action problems such as climate change or ocean pollution, several key factors that tend to increase actors’ likelihood of cooperation have been identified. These factors could also be applied to promote the rational use of antibiotics. They include a low level of anonymity for cooperation participants, a high level of public disclosure, communication among actors, the ability to punish undesirable behavior, and, notably, trust among the group of involved individuals [25].

Regarding practices, a high frequency of antibiotics being sold without a medical prescription was observed, primarily for urinary tract infections and upper respiratory tract infections. Among the reasons for selling antibiotics without a prescription, the difficulty patients face in obtaining a medical consultation is cited. In Colombia, according to the national assessment study of Health Promotion Entities services, the median waiting time for a general medical appointment is 4 days. Therefore, upon experiencing initial symptoms, patients turn to pharmacists for popular knowledge, seeking pharmaceutical recommendations to address their ailments [26]. Pharmacists, aware of the appointment waiting times, decide to prescribe medications without a medical prescription. While pharmacists bear responsibility for this inappropriate practice, the healthcare system has a greater responsibility to reduce waiting times for general medical care and to eliminate barriers hindering the population’s timely access to healthcare services, thereby discouraging unsafe practices.

This study identified three variables associated with practices: knowledge, attitudes, and type of pharmacy. The KAP model was developed in the 1950s, widely used in cross-sectional studies, and is grounded in the correlation between knowledge, attitudes, and practices under the assumption that enhancing personal knowledge will influence behavioral change [27]. However, there are several external factors that can shape practices beyond knowledge and attitudes. In this case, the type of pharmacy largely determined population behavior through two main pathways: (i) institutional rules imposed by chain pharmacies on their employees might lead them to modify their behavior out of fear of losing their jobs, and (ii) the fact that employees in chain pharmacies do not directly benefit from sales could prevent them from selling drugs without prescription.

This research comes with the following limitations: (i) the study has a cross-sectional design; hence, the associations do not imply causation; (ii) survey results might underestimate the sale of antibiotics without a medical prescription in the city due to social desirability bias and fear among pharmacists of sanctions from health authorities; and (iii) despite being the largest study conducted in the country, 47 pharmacies declined to participate in the research, which might impact the generalization of the findings.

### Conclusions

Pharmacists in Medellín exhibit inadequate knowledge, attitudes, and practices regarding the use and sale of antibiotics without a medical prescription. They ascribe curative properties to antibiotics for illnesses for which they are not indicated, lack confidence in the judgment of their colleagues for the rational use of antibiotics, believe that regulation would reduce their sales, and frequently sell antibiotics without a prescription for urinary tract and respiratory tract infections. These findings add to international evidence highlighting the necessity of designing educational and regulatory strategies to promote the rational use of these medications in pharmacies.

## 4. Materials and Methods

### 4.1. Study Population 

Based on a reference population of 2000 pharmaceutical establishments in the city of Medellín, an expected standard deviation of 10 in the KAP score, a confidence level of 95%, a design effect of 1, and a precision of 1, it was necessary to include 323 pharmacies. From this number, naturalistic pharmacies, establishments that were registered as pharmacies but were not operational at their listed addresses, and those that declined to participate were excluded. A total of 277 pharmacies were included, proportionally distributed across the city’s zones.

### 4.2. Data Collection Instrument 

The data collection instrument comprises 4 sections. The first section consists of 8 items that capture pharmacy characteristics and pharmacist details included in the research, such as pharmacy type (chain, retail) and location within the city, the number of antibiotics sold per day, and pharmacist demographics including gender, age group, education, and work experience. The second section assesses knowledge with 8 items using a 4-level Likert scale, where participants indicate their level of agreement from completely disagree to completely agree. The third section evaluates attitudes with 5 items using a 4-level Likert scale, where participants indicate their level of agreement or disagreement. The final section evaluates practices with 8 items, whereby participants indicate the frequency of performing the mentioned actions on a 4-level Likert scale ranging from never to always. (Appendix A).

The scale was designed to allow the presentation of both absolute and relative frequency of responses to each item and an aggregated score for each knowledge, attitude, and practice index. The score was transformed into a scale ranging from 0 (worst possible score) to 100 (best possible score), following the formula:$$\mathrm{Score}:\;\frac{Obtained\;score-lowest\;possible\;score}{Maximum\;possible\;score-minimum\;possible\;score}\times 100$$

This design enabled an analysis of responses in terms of absolute and relative frequencies and also allowed for a comprehensive assessment of knowledge, attitudes, and practices through calculated index scores.

### 4.3. Data Collection 

A pilot test was initially conducted with 10 pharmacists to ensure the acceptability of the instrument, question clarity, and avoid leading responses. Subsequently, the research team visited various pharmacies registered with the health department in the city. In each pharmacy, the project objectives were presented, and those agreeing to participate provided informed consent. Finally, the survey was self-administered and took approximately 15 to 20 min.

### 4.4. Data Analysis 

The data was analyzed using absolute and relative frequencies with 95% confidence intervals. Comparison of knowledge, attitudes, and practices scores was done using the median, interquartile range, Mann Whitney U-test, and Kruskal Wallis H-test, given the non-normality assumption as assessed by the Kolmogorov-Smirnov test with Lilliefors correction. Confounding variables were identified through linear regression models, evaluating assumptions of multicollinearity using the variance inflation factor, no autocorrelation of residuals using the Durbin-Watson test, linearity with ANOVA, normality of residuals, and homoscedasticity. Significance was considered for values of *p* < 0.05. All analyses were performed using SPSS V.29.

## Figures and Tables

**Figure 1 antibiotics-12-01456-f001:**
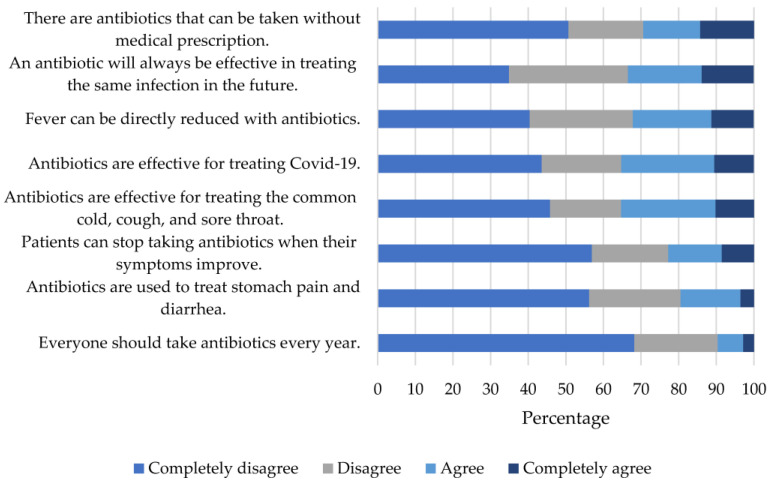
Relative Frequency of Responses to Items in the Knowledge Index.

**Figure 2 antibiotics-12-01456-f002:**
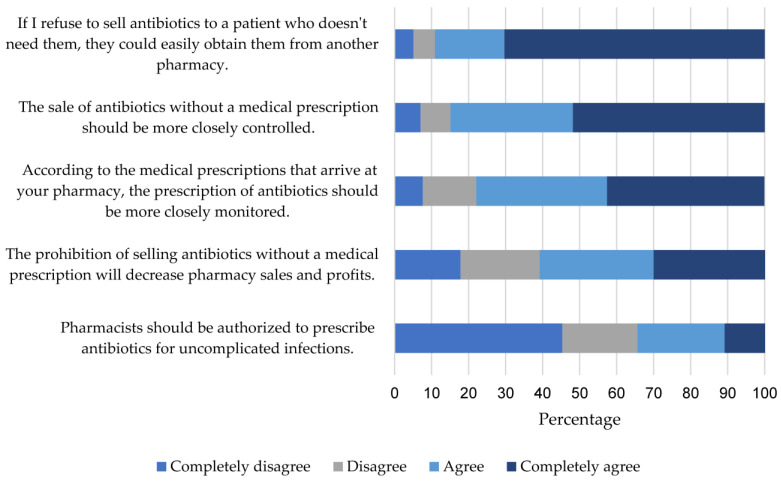
Relative Frequency of Responses to Items in the Attitudes Index.

**Figure 3 antibiotics-12-01456-f003:**
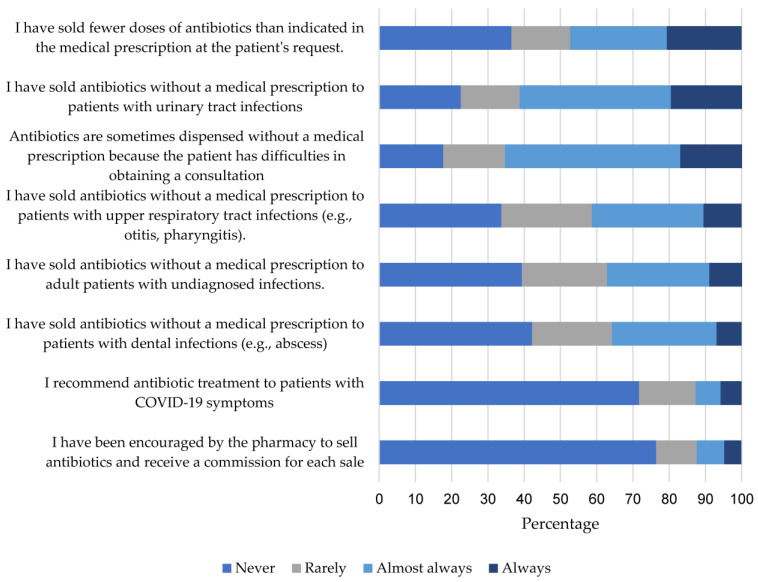
Relative Frequency of Responses to Items in the Practices Index.

**Figure 4 antibiotics-12-01456-f004:**
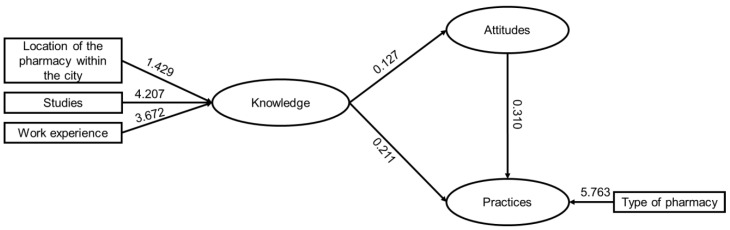
Regression Coefficients (β) of Factors Associated with Knowledge, Attitudes, and Practices in the Linear Regression Model. All coefficients have *p*-values < 0.05.

**Table 1 antibiotics-12-01456-t001:** Description of the characteristics of the pharmacies and the pharmacists included in the research.

		*n*	% (CI 95%)
Type of Pharmacy	Chain pharmacy	142	52.6 (46.6–58.5)
	Retail pharmacy	128	47.4 (41.5–53.4)
Location of the pharmacy within the city	Rural districts	5	1.8 (0.7–4.0)
	Northeastern zone	33	12.1 (8.6–16.3)
	Northwestern zone	40	14.7 (10.8–19.2)
	Eastern-central	60	22.0 (17.4–27.2)
	Western-central	58	21.2 (16.7–26.4)
	Southeastern zone	26	9.5 (6.5–13.4)
	Southwestern zone	51	18.7 (14.4–23.6)
Number of antibiotic tablets sold per day	<10	63	22.7 (18.1–27.9)
	10 to 30	91	32.9 (27.5–38.5)
	30 to 50	62	22.4 (17.8–27.6)
	>50	61	22.0 (17.4–27.2)
Age group	Youth (18–26 years old)	66	25.2 (20.2–30.7)
	Adult (27–59 years old)	174	66.4 (60.5–71.9)
	Older adult (>59 years old)	22	8.4 (5.5–12.2)
Sex	Women	171	61.7 (55.9–67.3)
	Man	106	38.3 (32.7–44.1)
Studies	Pharmacy Technician	111	40.1 (34.4–45.9)
	Pharmacy Assistant	134	48.4 (42.5–54.3)
	Other ^†^	32	11.6 (8.2–15.7)
Perception of education on antibiotics	<2 years	52	19.2 (14.8–24.2)
	2 to 5 years	59	21.8 (17.2–27.0)
	>5 years	160	59.0 (53.1–64.8)
Perception of education on antibiotics	Average to poor	27	9.9 (6.8–13.9)
	Good	182	66.9 (61.2–72.3)
	Excellent	63	23.2 (18.4–28.4)

Note: In some columns, the percentages do not add up to 277, due to missing values. ^†^ “Other” includes pharmaceutical chemist, dispenser, regency student, or those without any formal training.

**Table 2 antibiotics-12-01456-t002:** Comparison of the knowledge, attitudes, and practices based on the characteristics of the pharmacy and pharmacists.

		Knowledge Me (IQR)	Attitudes Me (IQR)	Practices Me (IQR)
Type of Pharmacy	Chain pharmacy	70.8 (58.3–87.5)	60.0 (46.7–66.7)	66.7 (54.2–79.2)
	Retail pharmacy	70.8 (54.2–87.5)	53.3 (40.0–63.3)	58.3 (45.8–75.0)
	** *p* ** **-value**	0.901	0.141	0.007 *
Location of the pharmacy within the city	Rural districts	62.5 (54.2–70.8)	60.0 (46.7–60.0)	58.3 (45.8–58.3)
	Northeastern zone	70.8 (62.5–83.3)	53.3 (40.0–66.7)	58.3 (45.8–75.0)
	Northwestern zone	83.3 (62.5–93.8)	53.3 (46.7–60.0)	62.5 (50.0–75.0)
	Eastern-central	75.0 (62.5–91.7)	53.3 (40.0–60.0)	66.7 (47.9–79.2)
	Western-central	79.2 (58.3–87.5)	53.3 (40.0–66.7)	64.6 (54.2–83.3)
	Southeastern zone	62.5 (45.8–70.8)	53.3 (40.0–66.7)	79.2 (62.5–87.5)
	Southwestern zone	66.7 (58.3–79.2)	60.0 (46.7–66.7)	58.3 (45.8–79.2)
	** *p* ** **-value**	0.004 *	0.784	0.081
Number of antibiotic tablets sold per day	<10	70.8 (62.5–87.5)	60.0 (46.7–66.7)	66.7 (45.8–83.3)
	10 to 30	75.0 (58.3–91.7)	60.0 (40.0–66.7)	62.5 (54.2–79.2)
	30 to 50	70.8 (62.5–83.3)	53.3 (40.0–66.7)	62.5 (50.0–79.2)
	>50	66.7 (50.0–87.5)	53.3 (40.0–60.0)	58.3 (45.8–79.2)
	** *p* ** **-value**	0.497	0.766	0.369
Age group	Youth (18–26 years old)	66.7 (54.2–83.3)	60.0 (40.0–66.7)	66.7 (54.2–83.3)
	Adult (27–59 years old)	75.0 (62.5–87.5)	60.0 (46.7–66.7)	66.7 (50.0–79.2)
	Older adult (>59 years old)	66.7 (50.0–87.5)	46.7 (40.0–53.3)	60.4 (45.8–83.3)
	** *p* ** **-value**	0.195	0.061	0.552
Sex	Women	70.8 (58.3–87.5)	53.3 (40.0–66.7)	66.7 (54.2–79.2)
	Men	70.8 (58.3–83.3)	60.0 (46.7–66.7)	58.3 (45.8–79.2)
	** *p* ** **-value**	0.398	0.171	0.163
Studies	Pharmacy Technician	79.2 (62.5–91.7)	53.3 (40.0–66.7)	66.7 (50.0–79.2)
	Pharmacy Assistant	66.7 (54.2–79.2)	60.0 (46.7–66.7)	62.5 (50.0–79.2)
	Other ^†^	75.0 (60.4–87.5)	53.3 (40.0–60.0)	58.3 (45.8–72.9)
	** *p* ** **-value**	0.001 *	0.151	0.378
Work experience	<2 years	66.7 (54.2–79.2)	56.7 (40.0–66.7)	66.7 (52.1–83.3)
	2 to 5 years	62.5 (54.2–83.3)	60.0 (40.0–66.7)	66.7 (50.0–79.2)
	>5 years	75.0 (62.5–91.7)	53.3 (46.7–66.7)	62.5 (47.9–79.2)
	** *p* ** **-value**	0.002 *	0.912	0.281
Perception of education on antibiotics	Average to poor	70.8 (58.3–87.5)	53.3 (40.0–60.0)	66.7 (54.2–75.0)
	Good	70.8 (58.3–87.5)	53.3 (46.7–66.7)	62.5 (50.0–79.2)
	Excellent	70.8 (50.0–83.3)	60.0 (46.7–66.7)	70.8 (54.2–83.3)
	** *p* ** **-value**	0.663	0.588	0.147

* *p*-value < 0.05 indicates statistically significant differences; ^†^ “Other” includes: pharmaceutical chemist, dispenser, regency student, or those without any formal training. The multiple comparisons, adjusted with the Bonferroni test, are provided in the Appendix A.

## Data Availability

Data has not been deposited in a public repository. Anonymized data is available on reasonable request to the authors.

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
