# Peer review of "Knowledge, Attitudes, and Practices Regarding Antibiotic Sales in Pharmacies in Medellín, Colombia 2023"

_antibiotics, 2023, doi:10.3390/antibiotics12091456_

Round 1

Reviewer 1 Report

Fascinating article about the knowledge, attitudes and practices of community pharmacists in Medellín.

Just a few minor comments.

The calculated sample size is 323. Does including only 270 affect type I and II errors?

In table 1. What is the role, and how is the confidence interval determined in nominal and ordinal variables?

In table 2. I recommend performing post hoc tests of the statistically significant results.

Author Response

Thank you for your guidance in reviewing our submission. The manuscript has been revised and the reviewers’ comments have been addressed below. We are thankful to the reviewers for their valuable suggestions for improving the manuscript, and we hope it is now acceptable for publication. Responses to each reviewer are included in yellow highlights within the manuscript

Reviewer 2 Report

The authors describe an important issue regarding the dispensing of antibiotics to patients without a prescription, along with attitudes and perceptions among pharmacists in that regard.

Minor grammatical errors.

Author Response

(The authors gave the same response as above.)

Reviewer 3 Report

The article entitled as Knowledge, Attitudes, and Practices Regarding Antibiotic Sales in Pharmacies in Medellín, Colombia 2023 by Daniel Ricardo Montes Colonia et al. This article is very interesting by describing the knowledge, attitudes, and practices regarding the sale of antibiotics in pharmacies in Medellín, Colombia. However, some minor issue needs rectification by the authors.

1. Add few lines about various classes of antibiotics commonly prescribing in Medellín, Colombia.

2.The questioner designed for the collections of data may please be attached as supplementary file.

3.Add limitations and future perceptive/recommendations of yours study.

Minor spelling errors needs rechecking before publishing 

Author Response

(The authors gave the same response as above.)

Reviewer 4 Report

This is an interesting study on Knowledge, Attitude and Practices (KAPs) regarding antibiotic sales in pharmacies in Medellin, since it covers a very important topic nowadays.

I appreciate the clarity, simplicity of the presentation. The figures and tables are representative. However, I have a few suggestions aiming to improve the manuscript quality:

1.      Table 1: “Number of antibiotic tablets sold per day” Really tablets? or boxes? Since an average of 5 days (5-10 tablets) are used for one therapy of any kind of infection

2.      Table 1: age groups-please specify in brackets the range of years for youth, adults and older

3.      table 1: studies-please specify who are others (pharmacists as well? pharmacist doctor? etc..)

4.      Table 2: the same specifications as above

5.      I missed the “Conclusion” at the and of the discussion, please highlight it as a sub-heading

Do you have any national legislation regarding antibiotic dispensing with or without prescription? How can pharmacist dispend antibiotic without prescription? Please specified this in the article.

I have no other comments, and after making these clarifications described above, I would accept it for publication.

The first sentence of the abstract shoul not be start with "to"

Author Response

(The authors gave the same response as above.)
